# BEYOND JOINT DEMONSTRATIONS: PERSONALIZED EXPERT GUIDANCE FOR EFFICIENT MULTI-AGENT REINFORCEMENT LEARNING

## ABSTRACT

Multi-Agent Reinforcement Learning (MARL) algorithms face the challenge of efficient exploration due to the exponential increase in the size of the joint state-action space. While demonstration-guided learning has proven beneficial in single-agent settings, its direct applicability to MARL is hindered by the practical difficulty of obtaining joint expert demonstrations. In this work, we introduce a novel concept of *personalized expert demonstrations* that an agent-specific expert provides. These demonstrations are tailored for an individual agent or, more broadly, for an individual *type* of agent in a heterogeneous team. It is crucial to emphasize that these demonstrations solely pertain to single-agent behaviors and do not encompass any cooperative elements. Consequently, it is essential to note that these demonstrations may not be inherently optimal when employed within a cooperative setting. To bootstrap the learning from the personalized expert demonstrations, we reformulate the MARL problem in occupancy measure space and propose two innovative algorithms, namely expert-guided MARL (EG-MARL) and Generalized EG-MARL (GEG-MARL). These algorithms involve the acquisition of *personalized reward signals* through demonstrations to guide agent exploration and the fostering of collaborative behaviors through environmental reward feedbacks. Our proposed algorithms are evaluated across both discrete and continuous environments. The results underscore the capacity of our methods to learn near-optimal policies even when provided with suboptimal demonstrations, and they excel in solving coordinated tasks that challenge state-of-the-art MARL algorithms.

## 1 INTRODUCTION

The use of expert demonstrations has been proven effective in accelerating learning in single-agent reinforcement learning, as evidenced by studies such as Kang et al. (2018); Chen & Xu (2022); Rengarajan et al. (2022). This approach has since been extended to Multi-Agent Reinforcement Learning (MARL) (Lee & Lee, 2019; Qiu et al., 2022), which typically assume the availability of high-quality collaborative joint demonstrations. However, from a practical standpoint, collecting joint demonstrations can be challenging. For example, consider a cooperative task (as shown in Figure 1) requiring two types of agents (green agents that open the doors and red agents that pick up the keys) to collect all the keys in an environment with several rooms. If human users provide expert demonstrations, we need one user per agent in a cooperative scenario. Furthermore, these demonstrations are also not scalable because if we change the number of robots or introduce new *types* of robots, we will need to gather a new set of demonstrations to learn from. In contrast, it is much easier to obtain demonstrations for individual robots, or even better, for each *types* of robot in a heterogeneous setting. For example, we can have one set of demonstrations for the red robots and one for the green robots. These demonstrations may not be optimal for the cooperative task. Therefore, there is a need to explore alternative methods that can facilitate efficient learning in multi-agent settings without relying on extensive expert demonstrations.

In this work, we introduce a novel notion of personalized expert demonstrations that an individual agent-specific expert provides. The personalized demonstration could be specific to an agent or specific to a type of agent in a heterogeneity team (Bettini et al., 2023). Since the personalized demonstrations will not necessarily reflect how the agents can cooperate, naively mimicking the

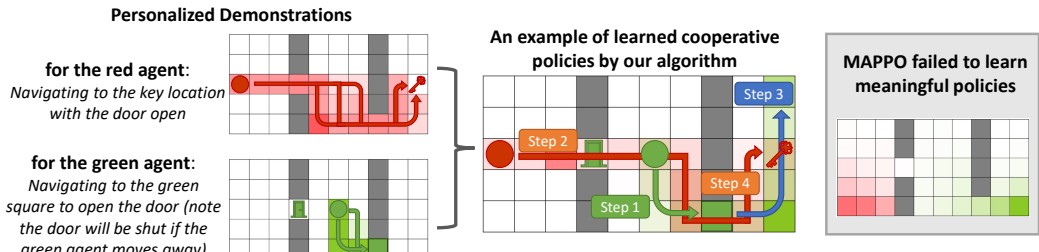

Figure 1: An example of utilizing personalized demonstrations to learn cooperative multi-agent policies. To learn successful cooperation, the agents are required not only to imitate the demonstrations (i.e., navigating to the corresponding goal location), but also to learn how to collaborate through environmental signal feedback (i.e., [step 1 and 2] the green agent stays on the green square to keep the door open and let the red agent enter the middle room, [step 3 and 4] the green agent moves away from the green square to allow the red agent enter the right room and navigate towards the key). We also visualize the state visitation frequency of the personalized demonstrations and the joint policies learned by our algorithm and MAPPO. The darker color means a higher value. We observe that the demonstrations guide the agents in exploring the state space more efficiently than in MAPPO.

demonstrations will not achieve cooperation. For example, the green agents may need to get out of the way to allow the red agents to pass through narrow corridors and collect the keys (see Figure 1). Therefore, purely imitation learning-based approaches would not be useful. We need an approach that uses personalized expert demonstrations as guidance and allows agents to learn to cooperate via collecting reward signals from the environments.

To this end, we present two decentralized MARL algorithms that learn from personalized expert demonstrations using a form of reward shaping to guide the exploration in the joint state-action space. The intuition is that in a sparse-reward environment, individual agents align their actions with their respective personalized demonstrations to fulfill their specific objectives in the absence of readily available reward signals. Concurrently, they adapt their behaviors and engage in cooperative efforts to attain shared objectives in response to infrequent collaborative feedback from the environment.

In the first method, termed **E**xpert **G**uided **M**ulti-**A**gent **R**einforcement **L**earning (**EG-MARL**), the reward-shaping component is calculated explicitly through occupancy measure matching. Here, occupancy measure denotes the probability of visiting each state and action pair under some given policy. We validate this algorithm on two discrete gridworld environments with varying numbers and types of agents, respectively, and demonstrate that it can boost exploration even with imperfect data. However, the occupancy measure calculation becomes quite expensive when the state space is large, and even impractical when the state and action space are continuous. Therefore, we present a generalized version of **EG-MARL**, termed **GEG-MARL**, that avoids estimating the occupancy measure for policy at each episode. It learns a discriminator for each agent to distinguish whether a local state-action pair is from its personalized demonstrations or the trained policy. The similarity scores produced by the discriminators are then used as implicit reward-shaping signals. We prove the effectiveness of this algorithm on both discrete gridworld and continuous multi-agent particle environments (Lowe et al., 2017; Mordatch & Abbeel, 2017), showing the EG-MARL framework is useful for practical applications. The main contributions of this paper are as follows:

(1) Our approach is the first to enable utilizing the personalized demonstrations for policy learning in heterogeneous MARL environments, regardless of the number and type of agents involved.

(2) We propose two algorithms (EG-MARL and GEG-MARL) that dynamically reshape the original reward to aid exploration. It is worth mentioning that both methods are general and compatible with most MARL policy gradient methods.

(3) We empirically validate the effectiveness of our methods with both optimal and suboptimal personalized demonstrations. Our approaches surpass state-of-the-art decentralized MARL algorithm, pure multi-agent imitation learning, and reward-shaping techniques in terms of performance. Furthermore, our methods demonstrate scalability in both heterogeneous and continuous environments.

## 2 RELATED WORKS

**Imitation Learning (IL).** IL algorithms aim to imitate an expert policy from demonstration data generated by that policy. Behavior Cloning (BC) is one of the most commonly used IL algorithms Pomerleau (1991); Bojarski et al. (2016), where the expert policy is estimated through supervised learning on demonstration data. However, BC suffers from compounding error caused by distribution shift Ross et al. (2011). Another thread of IL research is Inverse Reinforcement Learning (IRL) Ng et al. (2000); Ziebart et al. (2008), in which the underlying reward function is estimated from the demonstration data and used for policy learning. To alleviate the issue of long running time from IRL, Generative Adversarial Imitation Learning (GAIL) Ho & Ermon (2016); Song et al. (2018) has been proposed to directly obtain a policy from observed data while bypassing any intermediate IRL steps. In general, IL approaches can rarely perform better than demonstrations. Therefore, they are not directly suitable for scenarios where only personalized expert demonstrations that do not demonstrate how to collaborate are available.

**Learning from Demonstration (LfD).** Unlike IL approaches, the goal of LfD is to utilize demonstration data to aid learning instead of fully imitating. Existing LfD works exploit the demonstration data by adding them to a replay buffer with a prioritized replay mechanism to accelerate the learning process. Due to the off-policy nature of the demonstration data, most methods are value-based Hester et al. (2018); Vecerik et al. (2017). There has been some recent work where the demonstrations are used to aid exploration, especially in large state-action spaces Kang et al. (2018); Chen & Xu (2022); Rengarajan et al. (2022). For example, POfD Kang et al. (2018) learns an implicit reward from the demonstration data using a discriminator, and augments the original sparse reward with it; LOGO Rengarajan et al. (2022) uses the demonstration data to guide the policy update directly: during each update iteration, the algorithm tries to find a policy that is most similar to the behavior policy within a trust region. However, these works all focus on a single-agent setting.

In multi-agent settings, PwD-MARL Qiu et al. (2022) proposes to utilize demonstrations to pretrain agents by IL as a warm start, and then optimize the pretrained policies through normal MARL algorithms. Lee et al. Lee & Lee (2019) augments the experience buffer with the demonstration trajectories, and gradually reduces the degree of mixing demonstration samples during training to avoid the learned policy from being overly influenced by the demonstrations. These approaches, however, require joint demonstrations for all the agents which can be cumbersome (as we mentioned in section 1). The key difference in our approach is that we leverage only personalized expert demonstrations to learn a cooperative policy.

## 3 PROBLEM FORMULATION

We start by considering a Markov Decision Process $(\mathcal{S}, \mathcal{A}, \mathcal{P}, r, \gamma)$ where $\mathcal{S}$ denotes finite state space, $\mathcal{A}$ is finite action space, $\mathcal{P}$ denotes the transition function, $r$ represents the reward functions, and $\gamma \in (0, 1)$ is the discount factor. We consider a network of agents $\mathcal{G} = (\mathcal{V}, \mathcal{E})$, where $|\mathcal{V}| = N$ indicating the number of agents and $\mathcal{E}$ denotes the edge set for the network. An edge between two agents encodes their relationship by being neighbors and being able to communicate. Here, $\mathcal{S}$ denotes the global state across all the agents, which can be decomposed as the product of $N$ local spaces $\mathcal{S}_i$ as $\mathcal{S} = \mathcal{S}_1 \times \mathcal{S}_2 \times \cdots \times \mathcal{S}_N$. If the state of agent $i$ is $s_i \in \mathcal{S}_i, i \in \mathcal{V}$, then the global state is $s = (s_1, s_2, \cdots, s_N)$. Similarly, we define the global action space $\mathcal{A}$ as $\mathcal{A} = \mathcal{A}_1 \times \mathcal{A}_2 \times \cdots \times \mathcal{A}_N$, meaning that for any $a \in \mathcal{A}$, we may write $a = (a_1, a_2, \cdots, a_N)$ with $a_i \in \mathcal{A}_i, i \in \mathcal{V}$. Since we focus on decentralized learning, we define the global policy $\pi(a|s)$ as the product of local policies $\prod_{i=1}^{N} \pi^{(i)}(a_i|s)$. This is reasonable because this implies statistical independence between the agents in terms of taking actions. We extend this definition to a parameterized class of policies, $\pi_\theta(a|s)$ where $\theta \in \Theta$ are the parameters. Specifically, we have $\theta = (\theta_1, \theta_2, \cdots, \theta_N)$ as the factorized global policy parameters. Using policy factorization, we can write $\pi_\theta(a|s) = \prod_{i \in \mathcal{V}} \pi_{\theta_i}(a_i|s)$. We emphasize that the local policy parameters are kept private to each agent and are not allowed (or required) to be shared with other agents. For a given policy, we can write the global state action occupancy measure $\lambda^\pi(s, a)$ as

$$\lambda^\pi(s, a) = \sum_{t=0}^{\infty} \gamma^t \cdot \mathbb{P}\Big(s_t = s, a_t = a \ \Big| \ \pi, s_0 \sim \xi\Big) \tag{1}$$

and write the corresponding local cumulative state-action occupancy measure as

$$\lambda_i^\pi(s_i, a_i) = \sum_{t=0}^\infty \gamma^t \cdot \mathbb{P}\left(s_i^t = s_i, a_i^t = a_i \,\Big|\, \pi, s^0 \sim \xi\right) \tag{2}$$

for $\forall a_i \in \mathcal{A}_i, s_i \in \mathcal{S}_i$. An interesting observation to note here is that we can write the local occupancy measure as the marginalization of the global occupancy measure with respect to all other agents. Mathematically, it holds that

$$\lambda_i^\pi(s_i, a_i) = \sum_{a \in \{a_i\} \times \mathcal{A}_{-i}} \sum_{s \in \{s_i\} \times \mathcal{S}_{-i}} \lambda^\pi(s, a) \tag{3}$$

with $\mathcal{A}_{-i} = \Pi_{j \neq i} \mathcal{A}_j$ and $\mathcal{S}_{-i} = \Pi_{j \neq i} \mathcal{S}_j$. Utilizing the definition of local occupancy measure, we focus on the following multi-agent RL problem

$$\max_{\theta \in \Theta} R(\pi_\theta) := \frac{1}{N} \sum_{i=1}^N \langle \lambda_i^{\pi_\theta}, r_i \rangle. \tag{4}$$

We note that the problem in equation 4 is a generalization from single-agent RL to multi-agent RL, where $r_i$ are the local rewards for each agent. This problem denotes a cooperative multi-agent setting where we want to learn policy parameters that maximize the average of local rewards.

**MARL with Personalized Expert Demonstrations.** Now, we are ready to present the main problem we are interested in solving in this work. In single-agent settings, Kang et al. Kang et al. (2018) proposes the learning from demonstrations (LfD) framework, which estimates the JS-divergence between the occupancy measure from the expert and trained policy, and adds it to the standard reward objective as a regularization term. However, this approach is not directly applicable to multi-agent settings due to not knowing the occupancy measures of the joint state-action pairs from the expert demonstrations. As we discussed in Section 1, obtaining agent-specific personalized demonstrations is much easier compared to obtaining multi-agent joint demonstrations. With this intuition in mind, we are more interested in scenarios where individual-level demonstrations are accessible while joint demonstrations are hard to obtain.

Assume each agent is associated with a specific *type*, which can be represented as $E_i$. Here, a type refers to agents that are behaviorally identical, using the formalism from Bettini et al. (2023). We collect one set of expert demonstrations for each *type*. Agents with the same *type* also have the same local state space $\mathcal{S}_{E_i}$ and local action space $\mathcal{A}_{E_i}$. By letting an expert user perform each individual task in the corresponding local state and action space, we obtain a collection of expert demonstrations denoted by $\{\mathcal{B}_{E_i} = \{(s_{E_i}^t, a_{E_i}^t)\}_{t=0}^H\}$. We assume that the underlying expert policy associated with $\mathcal{B}_{E_i}$ is $\pi_{E_i}$, and $\lambda^{\pi_{E_i}}$ is the occupancy measure following the expert's policy $\pi_{E_i}$ for agent $i$.

We propose a version of learning from demonstration for multi-agent setting via maximizing the following objective

$$R(\pi_\theta) = F(\lambda^{\pi_\theta}) := \frac{1}{N} \sum_{i=1}^N F_i(\lambda_i^{\pi_\theta}) = \frac{1}{N} \sum_{i=1}^N \left( \langle \lambda_i^{\pi_\theta}, r_i \rangle - \eta D_{\text{JS}} \left( \lambda_i^{\pi_\theta} \,\|\, \lambda^{\pi_{E_i}} \right) \right), \tag{5}$$

where $\eta$ is a weighting term balancing between the long term reward and the policy similarity. It is worth mentioning that we can have different experts for different agents, or even different *types* of agents, which feature is not allowed by any existing algorithm in the literature.

## 4 EXPERT GUIDED MARL ALGORITHM

In this section, we derive our first algorithm **E**xpert **G**uided **M**ulti-**A**gent **R**einforcement **L**earning (**EG-MARL**). According to Zhang et al. (2021), we can define shadow reward $r^\pi(s, a) := \frac{\partial F(\lambda^\pi)}{\partial \lambda(s,a)}$, which can be optimized by the agents to solve the aforementioned objective instead of reward $r(s, a)$. However, $r^\pi(s, a)$ depends on the global knowledge of all local utilities and thus is not suitable for the decentralized setting. Therefore, we introduce the localized version of $r^\pi(s, a)$, the local shadow reward $r_i^\pi$ for each agent $i$:

$$r_i^\pi := \frac{\partial F_i(\lambda_i^{\pi_\theta})}{\partial \lambda_i(s_i, a_i)} = r_i - \eta \log \left( \frac{2 \cdot \lambda_i^{\pi_\theta}(s_i, a_i)}{\lambda_i^{\pi_\theta}(s_i, a_i) + \lambda^{\pi_{E_i}}(s_i, a_i)} \right) = r_i + \eta \bar{r}_i. \tag{6}$$

---

**Algorithm 1** **E**xpert **G**uided **M**ulti-**A**gent **R**einforcement **L**earning (EG-MARL)

---

**Input:** Agent number $N$; Initial parameters of policies and critics $\{\theta_i^0\}$ and $\{w_i^0\}$, where i = 1, 2, ..., N; Personalized expert trajectories $\{\mathcal{B}_{E_i} = \{(s_i^t, a_i^t)\}_{t=0}^H\}$ where i = 1, 2, ..., N; Batch size $\{B^k\}$; Episode length $\{H^k\}$; Weight parameters $\{\eta^k\}$.
**Output:** Learned policies $\{\pi_{\theta_i}\}$.

1: **for** agent $i = 1, 2, ..., N$ **do**
2:     Calculate empirical personalized expert state-action occupancy measures $\hat{\lambda}^{\pi_{E_i}}$ using equation 7.
3: **end for**
4: **for** iteration $k = 0, 1, 2, ...$ **do**
5:     Perform $B^k$ multi-agent Monte Carlo rollouts to obtain trajectories $\tau = \{(s^t, a^t)\}_{t=0}^{H^k}$ with initial distribution $\xi$, policy $\pi_{\theta^k}$ collected as batch $\mathcal{B}^k$.
6:     **for** agent $i = 1, 2, ..., N$ **do**
7:         Compute empirical local occupancy measure $\hat{\lambda}_i^{\pi_\theta}$ using equation 7.
8:         Estimate the reshaped rewards $\hat{r}_i^k$ using equation 6.
9:     **end for**
10:     Compute averaged reward across all agents.
11:     **for** agent $i = 1, 2, ..., N$ **do**
12:         Compute value estimates $V_i^t$ and advantage estimates $A_i^t$ based on the averaged reshaped rewards.
13:         Update $\theta_i$ by policy gradient: $\mathbb{E}_{\mathcal{B}^k} \left[ \nabla_{\theta_i} \pi_{\theta_i}(a_i|s) A_i(s, a_i) \right]$
14:         Update $w_i$ to minimize the mean-squared loss: $\mathbb{E}_{\mathcal{B}^k} \left[ (V_{w_i}(s) - V_i)^2 \right]$
15:     **end for**
16:     **for** agent $i = 1, 2, ..., N$ **do**
17:         Exchange information with neighbors $\mathcal{N}_i := \{j : (i, j) \in \mathcal{E}\}$:
    $w_i^{k+1} = \frac{1}{|\mathcal{N}_i|} \sum_{j \in \mathcal{N}_i} w_j^{k+1}$
18:     **end for**
19: **end for**

---

The first term $r_i$ is the original local reward, and the second term $\bar{r}_i$ can be viewed as an implicit personalized reward signal to augment the environmental exploration:

- When $\lambda_i^{\pi_\theta}(s_i, a_i) < \lambda^{\pi_{E_i}}(s_i, a_i)$, indicating the probability of occupying the current state-action pair is smaller than the expert's, we have $\bar{r}_i > 0$, a positive incentive is added to the original reward to encourage the visitation frequency;

- When $\lambda_i^{\pi_\theta}(s_i, a_i) = \lambda^{\pi_{E_i}}(s_i, a_i)$, we have $\bar{r}_i = 0$, the probability matches the expert one's, so the original reward is not reshaped;

- When $\lambda_i^{\pi_\theta}(s_i, a_i) > \lambda^{\pi_{E_i}}(s_i, a_i)$, we have $\bar{r}_i < 0$, a negative incentive is added to the original reward to decrease the visitation frequency.

The estimate of $r_i^\pi$ requires the calculation of personalized expert occupancy measures and local occupancy measures for each agent. In practice, we adopt Monte-Carlo rollouts to calculate them in a counting-based manner. Given a trajectory set $\mathcal{B} = \{(s^t, a^t)\}_{t=0}^H$, the empirical occupancy measure can be calculated as

$$\hat{\lambda} = \frac{1}{|\mathcal{B}|} \sum_{\tau \in \mathcal{B}} \sum_{t=0}^H \gamma^t \cdot e\left(s^t, a^t\right), \tag{7}$$

where $e(\cdot)$ is an indicator function. The personalized expert demonstrations are single-agent interactions with the environment, so we can use the above equation to calculate the associated occupancy measures directly. To calculate the local occupancy measure using multi-agent trajectories, we get the local state component from the global state, and replace $e\left(s^t, a^t\right)$ with $e\left(s_i^t, a_i^t\right)$ instead.

To solve the problem in a model-free manner, we proposed a decentralized actor-critic-based algorithm summarized in Algorithm 1. It is compatible with any policy gradient methods, *e.g.*, TRPO (Schulman et al., 2015) and PPO (Schulman et al., 2017). We adopted PPO-based actor updates in our implementation.

---

**Algorithm 2** **G**eneralized **E**xpert **G**uided **M**ulti-**A**gent **R**einforcement **L**earning (GEG-MARL)

---

**Input:** Agent number $N$; Initial parameters of policies, discriminators, and critics $\{\theta_i^0\}$, $\{\phi_i^0\}$, $\{w_i^0\}$, where i = 1, 2, ..., N; Personalized expert trajectories $\{\mathcal{B}_{E_i} = \{(s_i^t, a_i^t)\}_{t=0}^H\}$ where i = 1, 2, ..., N; Batch size $\{B^k\}$; Episode length $\{H^k\}$; Weight parameters $\{\eta^k\}$.
**Output:** Learned policies $\{\pi_{\theta_i}\}$.

1: **for** iteration $k = 0, 1, 2, ...$ **do**
2:   Perform $B^k$ multi-agent Monte Carlo rollouts to obtain trajectories $\tau = \{(s^t, a^t)\}_{t=0}^{H^k}$ with initial distribution $\xi$, policy $\pi_{\theta^k}$ collected as batch $\mathcal{B}^k$.
3:   **for** agent $i = 1, 2, ..., N$ **do**
4:     Update $\phi_i$ to increase the objective

$$\mathbb{E}_{\mathcal{B}^k}\left[\log D_{\phi_i}(s_i, a_i)\right] + \mathbb{E}_{\mathcal{B}_{E_i}}\left[\log(1 - D_{\phi_i}(s_i, a_i))\right] \tag{8}$$

5:     Estimate the reshaped reward as $\hat{r}_i^k = r_i - \eta^k \log D_{\phi_i}(s_i, a_i)$
6:   **end for**
7:   Step 10-18 in Algorithm 1.
8: **end for**

---

## 5 GENERALIZED EXPERT GUIDED MARL ALGORITHM

We note that the major bottleneck with Algorithm 1 is associated with the occupancy measure estimation. This step becomes quite expensive when the state space is large, making the algorithm ineffective, and even impractical when the state and action space are continuous. We deal with this issue by utilizing the idea from Ho & Ermon (2016), which replaces the JS divergence to the expert's demonstration with

$$D_{JS}(\lambda_\pi, \lambda_E) \approx \sup_D (\mathbb{E}_{\lambda_\pi}[\log(D(s,a))] + \mathbb{E}_{\lambda_E}[\log(1 - D(s,a))]), \tag{9}$$

where $D(s,a) : \mathcal{S} \times \mathcal{A} \longrightarrow \mathbb{R}$ is discriminative classifier. Next, we utilize the lower bound mentioned in equation 9 into equation 5, and rewrite the objective as

$$\min_\theta \sup_{D \in (0,1)} (-R(\pi_\theta, D)), \tag{10}$$

where objective $R(\pi_\theta, D)$ is given by

$$R(\pi_\theta, D) = \frac{1}{N}\sum_{i=1}^N \left[\langle\lambda_i^{\pi_\theta}, r_i\rangle + \beta H_i(\pi_{\theta_i}) - \eta\mathbb{E}_{\pi_{\theta_i}}[\log(D(s_i, a_i))] - \mathbb{E}_{\pi_{E_i}}[\log(1 - D(s_i, a_i))]\right]. \tag{11}$$

In equation 11, $H_i(\pi_{\theta_i})$ is for entropic regularization to avoid overfitting (Ziebart et al., 2008; Ziebart, 2010), and $\beta$ is a hyperparameter that weighs the regularization. We note that the problem in equation 10 is a minimax problem similar to the target in Generative Adversarial Networks (GANs) (Goodfellow et al., 2014). GANs aim to learn a generative model to produce samples that cannot be distinguished from the true distribution even with a well-trained discriminative model. In our case, the true distributions are the personalized expert policies $\pi_{E_i}(a_i|s_i)$, or equivalently the expert occupancy measures $\lambda^{\pi_{E_i}}$. Our approach is different from imitation learning in the sense that we only use personalized demonstrations as guidance rather than fully mimicking the expert demonstration. Suppose $D_i$ is parameterized by $\phi_i$. By labeling expert individual state-action pairs as true ("1") and policy local state-action pairs as false ("0"), we get the following objective:

$$\min_\theta \max_\phi \mathcal{L} = \frac{1}{N}\sum_{i=1}^N \left[-\langle\lambda_i^{\pi_\theta}, r_i\rangle - \beta H_i(\pi_{\theta_i}) + \eta(\mathbb{E}_{\pi_{\theta_i}}[\log(D_{\phi_i}(s_i, a_i))] + \mathbb{E}_{\pi_{E_i}}[\log(1 - D_{\phi_i}(s_i, a_i))])\right]. \tag{12}$$

Algorithm 2 presents the details of how we solve this optimization problem. We refer this algorithm as Generalized EG-MARL (GEG-MARL) to reflect the generalization ability provided by training a discriminator, rather than empirical estimation of local occupancy measures as in EG-MARL. As we show in the next section, GEG-MARL generalizes better to increasing number of agents as well as to continuous state-action spaces.

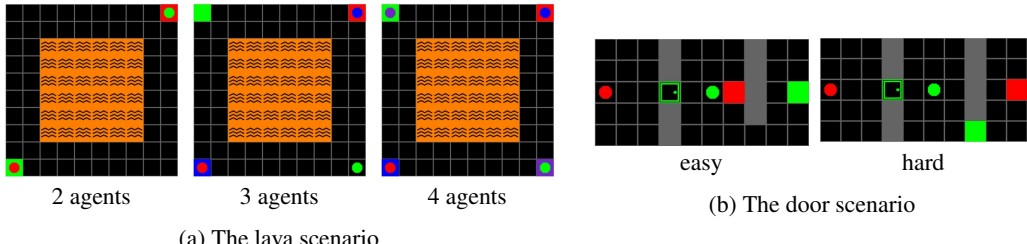

(a) The lava scenario       (b) The door scenario

Figure 2: The discrete gridworld environments. Circles indicate the start locations of the agents, and squares with the same color indicate the corresponding goal locations for each agent.

## 6 EXPERIMENTS

In this section, we investigate the performance of EG-MARL and GEG-MARL empirically. We focus on the following questions: (1) how do EG-MARL and GEG-MARL that leverage personalized demonstrations compare to state-of-the-art MARL techniques? (2) how does the sub-optimality of the personalized expert demonstrations affect the performance of EG-MARL and GEG-MARL? and (3) how do EG-MARL and GEG-MARL scale with an increasing number of agents and in case of continuous state-action spaces?

### 6.1 DISCRETE GRIDWORLD ENVIRONMENTS

We use two gridworld environments with discrete state and action space: the lava scenario and the door scenario (Figure 2). The agents in the lava scenario are homogeneous because they have the same objective: navigating to their corresponding goal location. The door scenario is heterogeneous: the assistant agent (green) must open the door while the other agent (red) must reach the goal.

- **The lava scenario:** This environment has a 6-by-6 lava pond in the center (Figure 2a). We provide three variants of this scenario, each involving differing quantities of agents and escalating levels of complexity. The primary objective of the agents is to expeditiously navigate to their designated goal locations while avoiding stepping into the lava. An episode terminates when all agents reach their respective goals (succeed), or if any agents step into the lava or the maximum episode length is reached (fail).

- **The door scenario:** This environment is adapted from Franzmeyer et al. (2022) (see Figure 2b). In this scenario, the green agent must navigate to a designated green square and maintain its presence there to sustain the open state of the green door, thereby enabling the entry of a red agent into the right side room. An episode ends when the red agent reaches the red goal location (succeed) or the maximum episode length is reached (fail).

Each agent's local state space is its $\{x, y\}$ coordinates in the map. We concatenate all the local states together to form the global state and assume all agents have access to the global state, which has a dimension of $\mathbb{R}^{n \times 2}$ ($n$ is the agent number). The local action space includes five actions: left, right, up, down, and stay. A sparse reward is granted when an episode succeeds, while a small penalty will be subtracted according to the steps taken ($10 - \text{step\_count}/\text{max\_step}$). Agents will receive a penalty of $-1$ if they collide with each other.

For each scenario, we collect two sets of personalized demonstrations for each s agent: *optimal* and *suboptimal*. The optimal demonstrations are collected by performing hand-crafted optimal policies for each individual task. The suboptimal demonstrations are collected with imperfect policies trained with PPO in the dense-reward setting through an early stopping, where the dense reward is naively defined as the negative distance between the agent and the goal. Examples of personalized demonstration are shown in Figure 1 for the door scenario and in Figure 7 (in Appendix B) for the lava scenario. The details of the suboptimal demonstrations are summarized in Table 1 (in Appendix B), the average episodic rewards of which are around 4.5 and about half of its optimal counterparts.

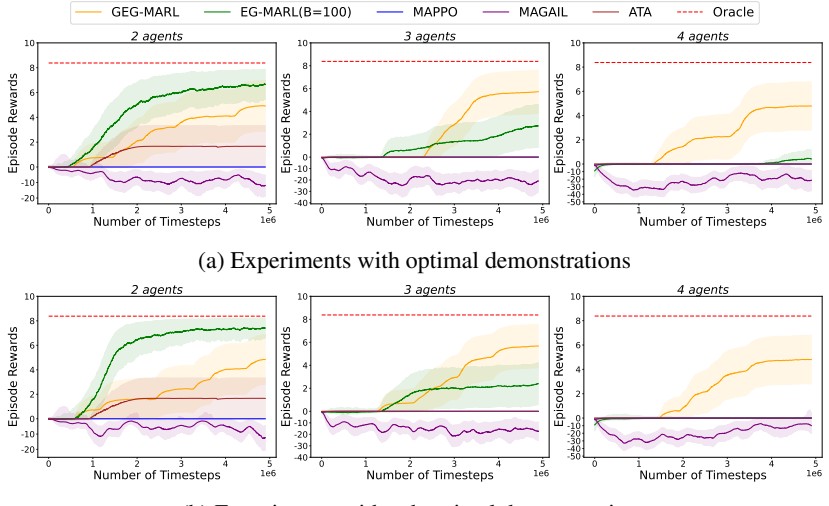

(a) Experiments with optimal demonstrations

(b) Experiments with suboptimal demonstrations

Figure 3: Learning curves of our algorithms versus other baseline methods under the **lava scenario**. The proposed EG-MARL and GEG-MARL algorithms converge to higher rewards than the baseline models. We can also observe that GEG-MARL generalizes better to larger numbers of agents.

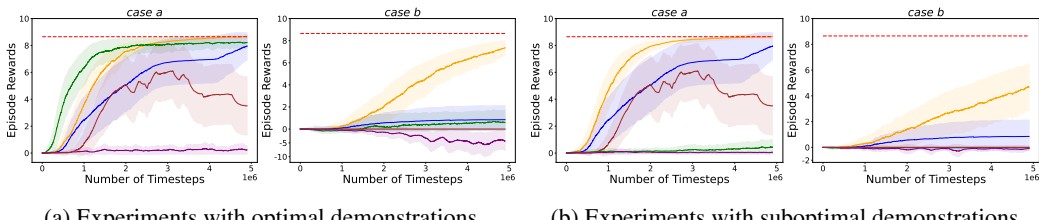

(a) Experiments with optimal demonstrations         (b) Experiments with suboptimal demonstrations

Figure 4: Learning curves of our algorithms versus other baseline methods under the **door scenario**. Our GEG-MARL algorithm shows better robustness in terms of convergence and generalizability.

## 6.2 COMPARISON WITH THE EXISTING METHODS

We compare EG-MARL (Algorithm 1) and GEG-MARL (Algorithm 2) against four strong baselines: (1) Oracle: the best return one can obtain with optimal policies; (2) MAPPO: training the policies with one of the best decentralized MARL algorithms, MAPPO Yu et al. (2021), in sparse-reward setting; (3) MAGAIL: training the policies with the state-of-art multi-agent imitation learning algorithm, MAGAIL Song et al. (2018), under the same setting as ours, where only individual demonstrations are provided; (4) ATA: training the policies with one of the best multi-agent reward-shaping methods, ATA She et al. (2022), in the sparse-reward setting. We run each method for each environment with 10 different random initializations and plot the mean and variance across all the runs.

With the lava scenario, we aim to illustrate the scalability of our algorithms across a spectrum of agent quantities. In contrast, the door scenario serves as a platform for showcasing the efficacy of our algorithm in a diverse and heterogeneous context. Furthermore, we conduct an assessment to ascertain the impact of suboptimal demonstrations on the performance of our algorithms.

**How do our algorithms scale with increasing number of agents?** The learning curves within the lava scenario are depicted in Figure 3. In the 2-agent scenario, EG-MARL emerges as the top-performing algorithm, demonstrating efficient learning. However, its performance exhibits a notable slowdown as the agent count escalates to 3 and 4, suggesting some limitations in scalability. In contrast, GEG-MARL exhibits superior generalizability across scenarios featuring a larger number of agents. It demonstrates an ability to navigate efficiently and maintain stable performance even when confronted with suboptimal demonstrations. In contrast, MAPPO struggles to acquire meaningful behavior across all scenarios, mainly due to the sparsity of rewards. MAGAIL performs even worse

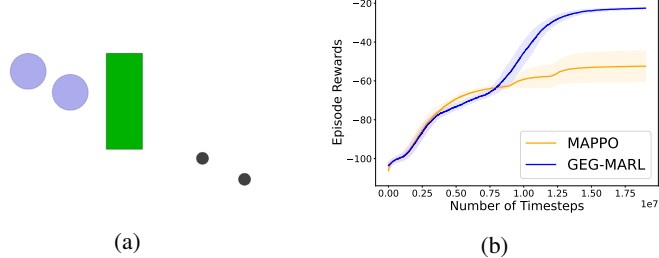

(a)                                                    (b)

Figure 5: (a) The modified cooperative navigation scenario. (b) Learning curves of GEG-MARL versus MAPPO under the modified cooperative navigation scenario.

than MAPPO, with agents frequently colliding due to the absence of environmental reward signals and reliance solely on individual agent demonstrations. ATA can acquire suboptimal policies in the 2-agent setting but experiences a decline in learning efficacy as the number of agents increases. This underscores the algorithm's limitations as the complexity of the multi-agent environment increases.

**How do our algorithms perform under the heterogeneous setting?** Figure 4 presents a comparative analysis of the learning curves for our algorithms and baseline methods within the door scenario. The easy case is more straightforward: the green agent must navigate to the green square and maintain its position there to keep the door open, thus allowing the passage of the red agent into the right side room and progressing toward its goal location. Most algorithms, with the exception of MAGAIL, demonstrate proficient performance in this context. Notably, EG-MARL and GEG-MARL exhibit the swiftest convergence. Intuitively, this scenario is so simple that success should be achievable by merely following individual demonstrations. MAGAIL's failure in this scenario arises from its inability to instruct the green agent to remain at the green square, a behavior not covered in the demonstrated actions. This further emphasizes the critical significance of environmental reward signals when integrating individual demonstrations into multi-agent learning paradigms. The hard case necessitates a higher degree of agent cooperation. Here, the green agent must initially open the door and maintain its position to allow the red agent access to the middle room. Subsequently, the green agent must disengage from the green square, thereby permitting the red agent to proceed into the right room and navigate toward its ultimate goal. In this complex setting, only GEG-MARL demonstrates commendable convergence. This can be attributed to the discriminator introducing increased variability in reward alterations, thereby fostering superior convergence dynamics.

**How does GEG-MARL perform in the continuous setting?** We modified the cooperative navigation task from the multi-agent particle environment Lowe et al. (2017); Mordatch & Abbeel (2017) to evaluate the performance of our algorithm in a continuous environment. EG-MARL cannot be used for this task since estimating occupancy measures through Monte Carlo rollouts in the continuous setting is not feasible. We do not compare with MAGAIL since simply imitating the experts is insufficient, as shown in the discrete environments' experiments. The modified Cooperative Navigation environment consists of 2 agents, 2 goal landmarks, and a wall between the agents and the goals. The agents need to navigate around the wall to occupy both landmarks. The state definition is the same as the original. We sparsify their original reward as follows: $r = \sum_{a \in A} \min(0.3, \min_{g \in G} d(a, g)) + \sum_{g \in G} \min(0.3, \min_{a \in A} d(a, g))$. Figure 5 demonstrates the learning curves of both algorithms. This validates the ability of GEG-MARL in continuous environments.

## 7    CONCLUSIONS

In this paper, we propose two algorithms for personalized expert-guided MARL. Unlike previous learning from demonstration approaches, we show how to incorporate expert demonstrations of only individual agents instead of joint demonstrations that are challenging to obtain for large teams in the real world. We demonstrate that our proposed algorithms, EG-MARL and GEG-MARL, outperform state-of-the-art imitation learning, learning-from-scratch and reward-shaping MARL algorithms. We further show that our algorithms are able to learn from suboptimal demonstrations and present the efficacy of GEG-MARL in continuous state-action spaces.

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

# A PROOF OF CONCEPT

We adopt the Pacman environment from Jiang & Lu (2021), as illustrated in Figure 6a, to prove the concept of personalized shadow reward. In this environment, four agents (yellow) are initialized in the maze center, and some red dots are randomly initialized in each room. The agents must explore the maze to eat as many dots as possible. Only a sparse reward equaling the total number of eaten dots is granted at the end of each episode.

It is apparent that the optimal strategy entails each agent taking responsibility for a specific room. In the absence of personalized guidance, MAPPO can easily acquire a suboptimal policy where multiple agents enter the same room, potentially leaving certain rooms unexplored (see Figure 6c). However, when individualized guidance is provided, such as an additional shadow reward of 0.1 for agent $i$ entering room $i$, the agents readily adapt to exploring distinct rooms (see Figure 6b), resulting in a substantially improved policy convergence (see Figure 6d). We use this example to illustrate the idea of personalized guidance. In the rest of the experiments reported, the shadow reward is not hand-crafted but is instead learned as given in EG-MARL and GEG-MARL.

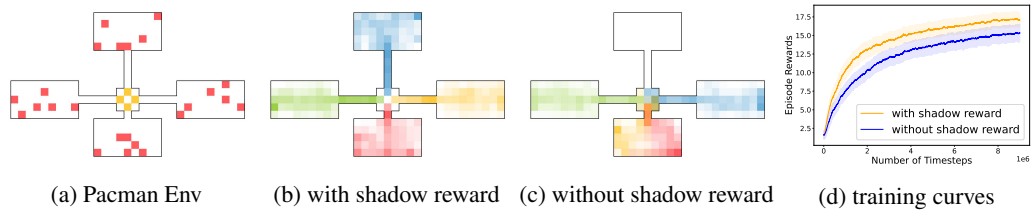

| (a) Pacman Env | (b) with shadow reward | (c) without shadow reward | (d) training curves |
|---|---|---|---|

Figure 6: (a) The Pacman environment. (b) and (c) Visualizations of state visitation frequencies, with different colors representing agents. (d) Training curve comparison.

# B ADDITIONAL EXPERIMENTS DETAILS AND VISUALIZATIONS

Figure 7 shows an example of personalized demonstrations for the lava scenario. Table 1 summarizes the details of the suboptimal demonstrations, whose average episodic rewards are around 4.5 and about half of their optimal counterparts.

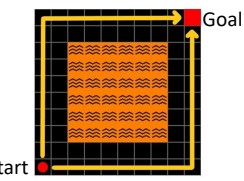

Figure 7: An example of personalized demonstrations for the lava scenario (we did not visualize all the optimal paths). There is only one agent in the environment. The agent may take both the left and the right path toward the goal.

| Agent Id | $\mathcal{S}$ | $\mathcal{A}$ | Samples | Average Episodic Reward |
|---|---|---|---|---|
| 1 | $\mathbb{R}^2$ | 5 | 300 | 4.42 |
| 2 | $\mathbb{R}^2$ | 5 | 440 | 4.79 |
| 3 | $\mathbb{R}^2$ | 5 | 344 | 4.3 |
| 4 | $\mathbb{R}^2$ | 5 | 360 | 4.41 |

(a) The lava scenario

| Case | Agent | $\mathcal{S}$ | $\mathcal{A}$ | Samples | Average Episodic Reward |
|---|---|---|---|---|---|
| easy | red | $\mathbb{R}^2$ | 5 | 643 | 4.01 |
| easy | green | $\mathbb{R}^2$ | 5 | 612 | 4.39 |
| hard | red | $\mathbb{R}^2$ | 5 | 607 | 4.34 |
| hard | green | $\mathbb{R}^2$ | 5 | 593 | 4.36 |

(b) The door scenario

Table 1: The details of suboptimal demonstrations.

We provide visual representations of the policy and state occupancy measure corresponding to suboptimal demonstrations in Figure 8 for the lava scenarios and in Figure 9 for the door scenario. The red square symbolizes the agent's initial position in these visualizations, while the green square designates its respective goal location. Arrows within the figures denote available actions at each state, with arrow length indicating the probability associated with each action.

We additionally depict the state visitation frequencies of the joint policies learned by GEG-MARL with suboptimal demonstrations and MAPPO for both the lava (Figure 10) and the door scenario (Figure 11).

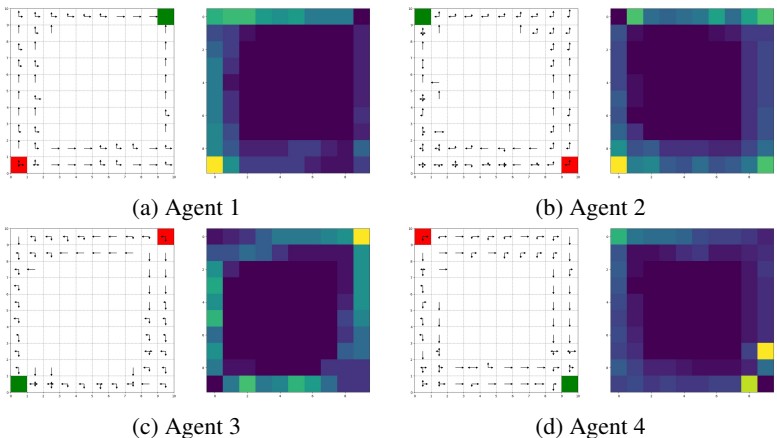

(a) Agent 1      (b) Agent 2

(c) Agent 3      (d) Agent 4

Figure 8: Agent policies and state occupancy measures estimated from the suboptimal demonstrations for the **lava scenario**.

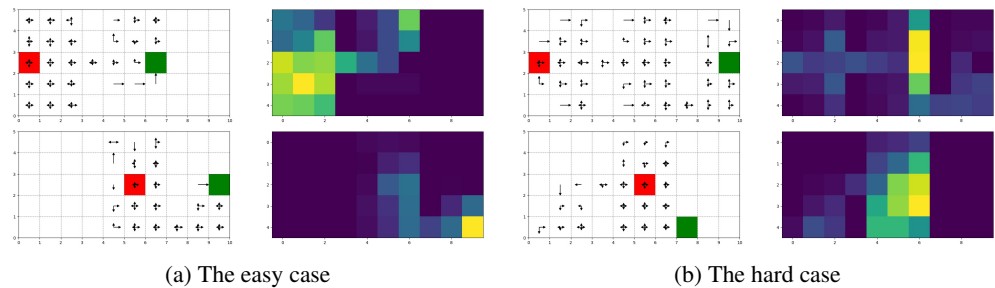

(a) The easy case      (b) The hard case

Figure 9: Agent policies and state occupancy measures estimated from the suboptimal demonstrations for the **door scenario**. The top row is for the red agent, and the bottom row is for the green agent.

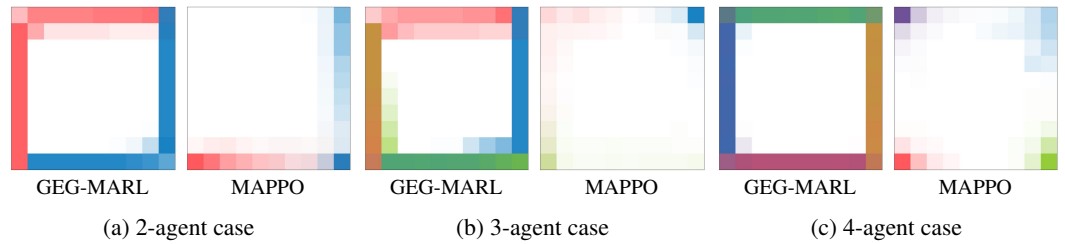

GEG-MARL    MAPPO      GEG-MARL    MAPPO      GEG-MARL    MAPPO

(a) 2-agent case      (b) 3-agent case      (c) 4-agent case

Figure 10: State visitation frequency of the joint policies learned by GEG-MARL (with suboptimal demonstrations) and MAPPO for the **lava scenario**. The darker color means a higher value. MAPPO failed to learn any meaningful policies in all three settings.

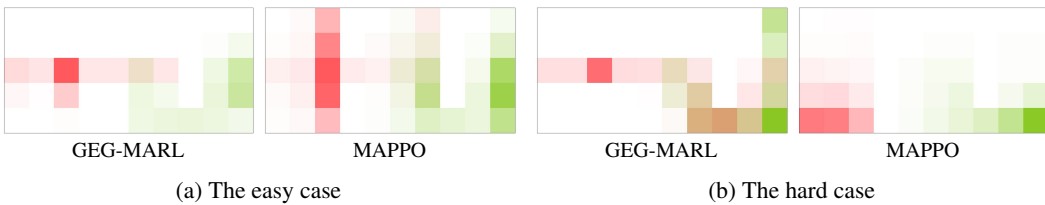

GEG-MARL      MAPPO      GEG-MARL      MAPPO

(a) The easy case      (b) The hard case

Figure 11: State visitation frequency of the joint policies learned by GEG-MARL (with suboptimal demonstrations) and MAPPO for the **door scenario**. The darker color means a higher value. MAPPO converges to a suboptimal policy in the easy case, while fails to learn in the hard case.

## C  THE INFLUENCE OF BATCH SIZE $B$ IN OCCUPANCY MEASURE ESTIMATION

We also investigate the impact of the batch size $B$ on empirical occupancy measure estimation in Algorithm 1. We conduct a comparative analysis across three settings: $B = 100, 1000, 10000$. Initially, there was an anticipation that an increased batch size would expedite policy convergence and result in higher returns due to more accurate occupancy measure estimation. However, as observed in the experimental results presented in Figure 12, this hypothesis does not hold true. Interestingly, the learning curves demonstrate that policy convergence rates remain relatively consistent across various values of $B$.

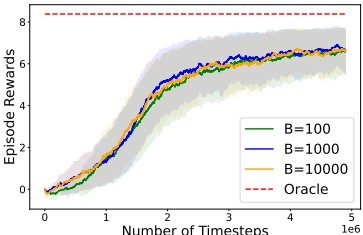

Figure 12: lava (2-agent case) environment with varying numbers of samples for computing the occupancy measure.

