# OpenReview forum: "Beyond Joint Demonstrations: Personalized Expert Guidance for Efficient Multi-Agent Reinforcement Learning"
_ICLR.cc/2024/Conference — ICLR 2024 Conference Withdrawn Submission_

### Official Review · Reviewer_oHLh · 2023-10-16

**Soundness:** 2 fair
**Presentation:** 3 good
**Contribution:** 2 fair
**Rating:** 3
**Confidence:** 3

**Summary:**

This paper aims to incorporate imitation learning (IL) as guidance for multi-agent reinforcement learning (MARL) to achieve better performance. The demonstration for IL of each type of agents is "personalized", i.e., may not be global optimal. The learning objective is thus maximizing the expected return over all agents while minimizing the discrepancy between agent policy's occupancy measure and the personalized demonstration's occupancy measure. Experiment results show better performance compared with baselines.

**Strengths:**

Originality: the paper claims it is the first to utilize personalized demonstrations for heterogeneous MARL environments.

Clarity: the presentation of the paper is easy to follow.

**Weaknesses:**

1. The idea of adopting "personalized" or non-global optimal demonstration as guidance for MARL is not new, e.g., in robotics research filed. Some examples are [1] [2].

2. Prevent exploration: Since part of the learning objective is to minimize the discrepancy between occupancy measures, it can naturally prevent the exploration due to this occupancy measure constraint.

3. Hinder policy learning: Though not formally defined, my understanding of this personalized demonstration is that it is non-global optimal. Then a large weight of imitation (large $\eta$) may hinder the policy learning in cooperative case. For example, in Section 1, authors provide an example where two agents need to cooperate to finish a task while the personalized demonstrations do not reflect the cooperation. A large $\eta$ may prevent the two agents to cooperate. I did not find the discussion of the choice of $\eta$ in the paper.

4. Some claims in the paper are questionable. e.g.,  in page 4, in paragraph MARL with Personalized Expert Demonstrations, "obtaining agent-specific personalized demonstrations is much easier compared to obtaining multi-agent joint demonstrations". This is not really the case, e.g., in mixed cooperative-competitive case, it is not easy to get personalized demonstrations without knowing the actions from other agents. If this paper only focuses on the case "where individual-level demonstrations are accessible", the
scenarios are quite limited.

5. The test environments and the agent size are small, hard to justify the method itself.

[1] MAPPER: Multi-Agent Path Planning with Evolutionary Reinforcement Learning in Mixed Dynamic Environments, IROS 2020

[2] Mobile Robot Path Planning in Dynamic Environments Through Globally Guided Reinforcement Learning, RA-L

**Questions:**

1. Personalized demonstration is the core idea of this paper but lacks a formal definition. Could you define formally when demonstration is regarded as "personalized"?

2. How to determine the $\eta$ in your algorithm?

3. Could you explain why MAPPO performs poorly in your environments? A natural reason may be your algorithm has extra guidance while others do not. Then it is more reasonable to compare with other exploration MARL methods.

---

> ### Author Response · Authors · 2023-11-22
>
> We thank all the reviewers for their valuable comments. We will incorporate their constructive suggestions to enhance the quality of our work.

---

### Official Review · Reviewer_G5rt · 2023-10-23

**Soundness:** 3 good
**Presentation:** 3 good
**Contribution:** 3 good
**Rating:** 6
**Confidence:** 3

**Summary:**

The authors investigate sparse rewards in multi-agent reinforcement learning and propose using per-agent demonstrations to help solve the exploration problem. They introduce EG-MARL, an algorithm which enforces similar behaviour to the demonstration policy by adding an additional reward term comparing the state-action occupancies of the two policies.

To address the lack of scalability in this approach, the authors also introduce GEG-MARL, which replaces the occupancy measure with a learned discriminator. They then compare their algorithm to a number of baselines on two sparse-reward gridworlds and a modified MPE environment.

**Strengths:**

- The writing of the paper is clear.
- The setting is novel to my knowledge and well-motivated -- there may be settings where individual demonstrations are easy to come by, but joint trajectories would require significantly more work.
- The authors compare to a number of relevant baselines and evaluate on a reasonable number of different environments.

**Weaknesses:**

- All the evaluation is either on MPE or grid worlds. This is a significant limitation because these are relatively simple environments. It would be interesting to understand whether personal demonstrations can help in settings where sparse reward is not necessary, but exploration is still difficult. SMAC has a sparse reward setting -- it would be interesting to see how this approach fares across a selection of SMAC maps.
- The evaluation only focuses on a small number of agents, with a maximum of four. This is another limitation of the evaluation of GEG-MARL. As the number of agents increases, I would expect that the number of ways for agents to interact, and therefore for the personal demonstrations to be misleading, to increase. It would be interesting to see how GEG-MARL handles this setting.

**Questions:**

- I was slightly confused by the description of the setting. How is the setting different from a Dec-MDP [1]?
- What communication was used in the tasks? The communication graph is mentioned but I couldn't find details of the communication in the experiments.

[1] A Concise Introduction to Dec-POMDPs. Olihoek and Amato. https://www.fransoliehoek.net/docs/OliehoekAmato16book.pdf

---

> ### Author Response · Authors · 2023-11-22
>
> We thank all the reviewers for their valuable comments. We will incorporate their constructive suggestions to enhance the quality of our work.

---

### Official Review · Reviewer_C51y · 2023-10-26

**Soundness:** 1 poor
**Presentation:** 2 fair
**Contribution:** 2 fair
**Rating:** 3
**Confidence:** 4

**Summary:**

This paper aims to leverage demonstrations to bootstrap multi-agent reinforcement learning (MARL). Collecting joint demonstrations is expensive and individual demonstrations would lead to sub-optimal policies and can't learn to cooperate. This paper proposes personalized expert demonstrations that can guide agent exploration and cooperation. This paper leverages occupancy measure space and GAIL that are usually used in Learning from Demonstration (LfD) of single agent.

**Strengths:**

* This paper tries to address the issues of the joint demonstrations and individual demonstrations and proposes personalized expert demonstrations. The approach can show great effect on both homogeneous and heterogeneous settings. The idea is simple and direct.
* This paper leverages occupancy measure space and GAIL that are useful in single-agent settings and outperform all baselines from suboptimal demonstrations.

**Weaknesses:**

* The experiments are too toy. The largest number of agents is 4. It's not convincing that the approach can generalized to more numbers.
* The paper misses the explanation about how to collect personalized expert demonstrations or how to determine which type of each agent.
* Miss of ablation studies.

**Questions:**

* What's the performance if the number of agents increases (e.g. 10)?
* What's the performance of the more complex environment (e.g. Google football or starcraft)?
* How to collect personalized expert demonstrations or how to determine which type of each agent?
* It's wired that MAPPO and MAGAIL can't work in lava of 2 agents. Despite its sparse reward, it's still very simple in a 2-agent setting. Can you explain that?
* There should be more analysis. Like what's the effect of each part of the method(i.e. personalized demonstrations and GAIL)?
* There should be more baselines. A simple one is MAPPO with individual demonstrations or personalized demonstrations as a warm start.

---

> ### Author Response · Authors · 2023-11-22
>
> We thank all the reviewers for their valuable comments. We will incorporate their constructive suggestions to enhance the quality of our work.

---

### Official Review · Reviewer_eCAT · 2023-10-31

**Soundness:** 1 poor
**Presentation:** 2 fair
**Contribution:** 1 poor
**Rating:** 3
**Confidence:** 4

**Summary:**

This paper introduces a multiagent reinforcement learning (MARL) algorithm guided by demonstrations of single-agent expert policies to enhance learning efficiency. The proposed method, called EG-MARL, learns policies based on the value function and the Jensen-Shannon (JS) distance between the occupancy measure of the current agent's policy and that of the expert. In cases of continuous state-action spaces where empirical estimation of occupancy measures is challenging, the authors propose GEG-MARL, which uses a GAN-based discriminator. GEG-MARL outperforms other MARL baselines and exhibits better scalability in terms of state-action space and the number of agents

**Strengths:**

1. This paper is well-written with good readability and comprehensibility. It illustrates a motivating example, making it easy to understand the objective of this paper and the proposed method.

**Weaknesses:**

1. The theory presented in this paper appears to have issues. The joint value function is decomposed into value functions for each agent in eq (4), but this decomposition does not necessarily hold in general. The conditions, such as IGM, for successfully decomposing a joint value function have been widely studied and value decomposition can only be achieved when such conditions are met. Therefore, to successfully justify the decomposition of a joint value function, the authors should provide a condition and prove that the decomposition holds under that condition.
2. It has been well-studied in many previous works that generating intrinsic rewards based on a GAN discriminator, and applying such intrinsic rewards solely for learning from demonstrations, lacks novelty. This would be more novel if it addressed a specific problem that arises in multiagent environments, which does not appear to be the case here.
3. Comparing EG-MARL and GEG-MARL with MAPPO and MAGAIL seems unfair. The reason for this is that MAPPO does not use expert demonstrations, while MAGAIL does not utilize information obtained through interaction with the environment. Since there are several existing works on learning from demonstrations in a single-agent setting, it would be more appropriate to compare the proposed methods with simple multiagent extensions of such methods.

**Questions:**

1. [About Weakness 1] Could you kindly provide the proof of eq (4) and the conditions under which it holds?
2. [About Weakness 2 and 3] It appears that each agent learns its policy independently using eq (5). How does this method differ from a straightforward extension of a single-agent learning from demonstration method to a multiagent setting?

---

> ### Author Response · Authors · 2023-11-22
>
> We thank all the reviewers for their valuable comments. We will incorporate their constructive suggestions to enhance the quality of our work.